# Increased prevalence of transfusion-transmitted diseases among people with tattoos: A systematic review and meta-analysis

Sung Ha Lim[1◉], Solam Lee[1,2◉], Young Bin Lee[1], Chung Hyeok Lee[1], Jong Won Lee[1], Sang-Hoon Lee[ID][1], Ju Yeong Lee[1], Joung Soo Kim[3], Mi Youn Park[4], Sang Baek Koh[2], Eung Ho Choi[ID][1]*

1 Department of Dermatology, Yonsei University Wonju College of Medicine, Wonju, Republic of Korea,
2 Department of Preventive Medicine, Yonsei University Wonju College of Medicine, Wonju, Republic of Korea, 3 Department of Dermatology, Hanyang University College of Medicine, Guri, Republic of Korea, 4 Department of Dermatology, National Medical Center, Seoul, Republic of Korea

◉ These authors contributed equally to this work.
* choieh@yonsei.ac.kr

**Data Availability Statement:** All relevant data are within the paper and its Supporting Information files.

## Abstract

Whether having a tattoo increases the risk of transfusion-transmitted diseases (TTDs) is controversial. Although a few studies have suggested a strong association between having tattoos and TTDs, other studies have not shown the significance of the association. In addition, previous studies mainly focused only on hepatitis C viral infections. The objective of our study was to identify the prevalence and risk of TTDs in people with tattoos as compared with the non-tattooed population. A systematic review of the studies published before January 22, 2021, was performed using the Pubmed, Embase, and Web of Science databases. Observational studies on hepatitis C virus (HCV), hepatitis B virus (HBV), human immunodeficiency virus (HIV), and syphilis infections in people with and without tattoos were included. Studies that reported disease status without serological confirmation were excluded. A total of 121 studies were quantitatively analyzed. HCV (odds ratio [OR], 2.37; 95% confidence interval [CI], 2.04–2.76), HBV (OR, 1.55; 95% CI, 1.31–1.83), and HIV infections (OR, 3.55; 95% CI, 2.34–5.39) were more prevalent in the tattooed population. In subgroup analyses, the prevalence of HCV infection was significantly elevated in the general population, hospital patient, blood donor, intravenous (IV) drug user, and prisoner groups. IV drug users and prisoners showed high prevalence rates of HBV infection. The prevalence of HIV infection was significantly increased in the general population and prisoner groups. Having a tattoo is associated with an increased prevalence of TTDs. Our approach clarifies in-depth and supports a guideline for TTD screening in the tattooed population.

**Funding:** EHC was supported by the Korean Dermatology Research Foundation (2019). The funder had no role in study design, data collection and analysis, decision to publish, or preparation of the manuscript.

**Competing interests:** The authors have declared that no competing interests exist.

## Introduction

Tattoos are becoming rapidly popular among young people, as they have become recognized as a means of self-expression [1, 2]. According to a worldwide survey conducted in 2019 [3], the prevalence of tattoos was reported to range from 12.2% to 31.5% depending on the region. A population-based study found that more than one-third of young adults in the United States have tattoos [4]. Tattooing is an invasive procedure that involves the injection of pigmentary particles into the dermal layer of the skin through repeated skin punctures. Therefore, it poses a potential risk of infection by diverse microorganisms if the ink or instrument used for tattooing is reused without a proper disinfection procedure.

Transfusion-transmitted diseases (TTDs) are blood-borne infectious diseases that include hepatitis C virus (HCV), hepatitis B virus (HBV), human immunodeficiency virus (HIV), and syphilis infections [5, 6]. In 2016, 1.90 million patients with HCV infection in the United States were identified, but only 49.8% of them knew about their disease status [7]. Moreover, 1.10 million patients with HIV infection were also reported, but one-seventh of them were not aware of their infection [8]. The most important risk factor for the transmission of TTDs has been known as sharing needles and equipment for drug use.

However, whether having tattoos increases the risk of transmission of TTDs is still controversial. A few studies have suggested a strong association between having tattoos and TTDs [9–13]; however, other studies have not shown the significance of the association [14–16]. Some systematic reviews and meta-analyses [17–20] have reported that tattooing in certain groups could increase the risk of HCV transmission. However, previous studies were mainly limited to HCV infection, and studies performing comprehensive evidence syntheses, including other TTDs, using a uniform methodology are currently lacking. Therefore, the purpose of this systematic review and meta-analysis was to investigate the prevalence and risk of HCV, HBV, HIV, and syphilis infections in people with tattoos as compared with the non-tattooed population.

## Materials and methods

### Search strategy

We performed a comprehensive literature search in accordance with the Preferred Reporting Items for Systematic Reviews and Meta-Analyses (PRISMA) reporting guidelines. The Pubmed, Embase, and Web of Science databases were searched. One of the main reviewers (S. L.) performed the literature search, using the following search keywords: "tattoo*," "HIV," "AIDS," "immunodef*," "hepatitis," "HCV," "HBV," "HBsAg," "syphilis," "VDRL," "TPHA," "treponema*," "transfu*," "blood*," and "infect*." The detailed search strategies for the databases are summarized in S1 File. The literature search included the studies published until January 22, 2021. Articles written in English and Korean were included because of the authors' proficiencies in these languages.

### Study selection

Four main reviewers (S.L., S.H.L., Y.B.L., and C.H.L.) evaluated the titles and abstracts of the retrieved studies. All individual studies were independently reviewed by at least two reviewers. Any disagreements between the reviewers regarding the suitability of the studies were discussed with two other reviewers (J.W.L. and S.H.L.) and resolved by consensus. All the observational studies on HCV, HBV, HIV, and syphilis infections that investigated both individuals with and without tattoos were included. Meanwhile, the following studies were excluded: 1) non-research articles, 2) studies that reported disease status without serological confirmation

(e.g., self-response questionnaire), 3) studies that only investigated either subjects with or without tattoos (non-comparability), and 4) studies with insufficient sample sizes (n < 20).

## Data extraction and quality assessment

Data regarding publication details, study setting, population demographics, and serological findings were extracted from each study. The number of events (positive in serology) and total observations for HCV, HBV, HIV, and syphilis infections in both the tattooed and non-tattooed groups were extracted from case-control, cross-sectional, and case-series studies. For cohort and any other studies that performed a time-to-outcome analysis, the hazard ratio (HR) for each finding was directly extracted. The adapted Newcastle-Ottawa scale for assessing the quality of observational studies was used for the assessment of the analyzed studies [21]. Finally, the articles with adequate quality (score ≥ 3) were included in the quantitative meta-analysis.

## Data synthesis and outcomes

For case-control, cross-sectional, and case-series studies, we calculated the odds ratio (OR) as a summary statistical variable for comparing the prevalence of TTDs in the tattooed and non-tattooed populations. The HRs obtained from each study were to be meta-analyzed with time-to-outcome analysis. A random-effects model was used in the meta-analysis because a significant heterogeneity between the included studies was expected. The $I^2$ statistics was used to estimate and quantify the heterogeneity between the studies. Subgroup analyses (general population, hospital patients, blood donors, intravenous (IV) drug users, and prisoners) for the study populations (≥3 studies) were performed to address the heterogeneity. General population were those who were included population-based (community) studies. In contrast to other subgroups, which are within characteristic environments that favor an increased prevalence of TTDs in the tattooed population, we thought it is worth estimating the prevalence in a sample which can be approximated to the general population. Egger's linear regression test was performed to evaluate publication bias. The trim-and-fill method was used to adjust the summary statistics when a significant publication bias was detected. The analysis was performed using R version 3.5.0 (R Foundation for Statistical Computing). A $p$ value of <0.05 was considered statistically significant.

## Code and data availability

The final datasheet for extraction, program code used for the quantitative synthesis, and forest plots for individual analyses can be accessed at our public repository at https://doi.org/10.17632/cgpp4fzxhd.1.

## Results

### Study selection and characteristics

The PRISMA flow diagram is presented in Fig 1. Among the 2,030 publications screened, 798 full-text articles were assessed for eligibility. A total of 121 studies were quantitatively analyzed (Tables 1 and S1 and S2). However, none was a cohort study that included a time-to-outcome analysis. The adapted Newcastle-Ottawa scale for assessing the quality of the included cross-sectional studies is presented in S3 Table. The summary statistics for the HCV, HBV, HIV, and syphilis infections in the tattooed group as compared with the non-tattooed group are shown in Fig 2.

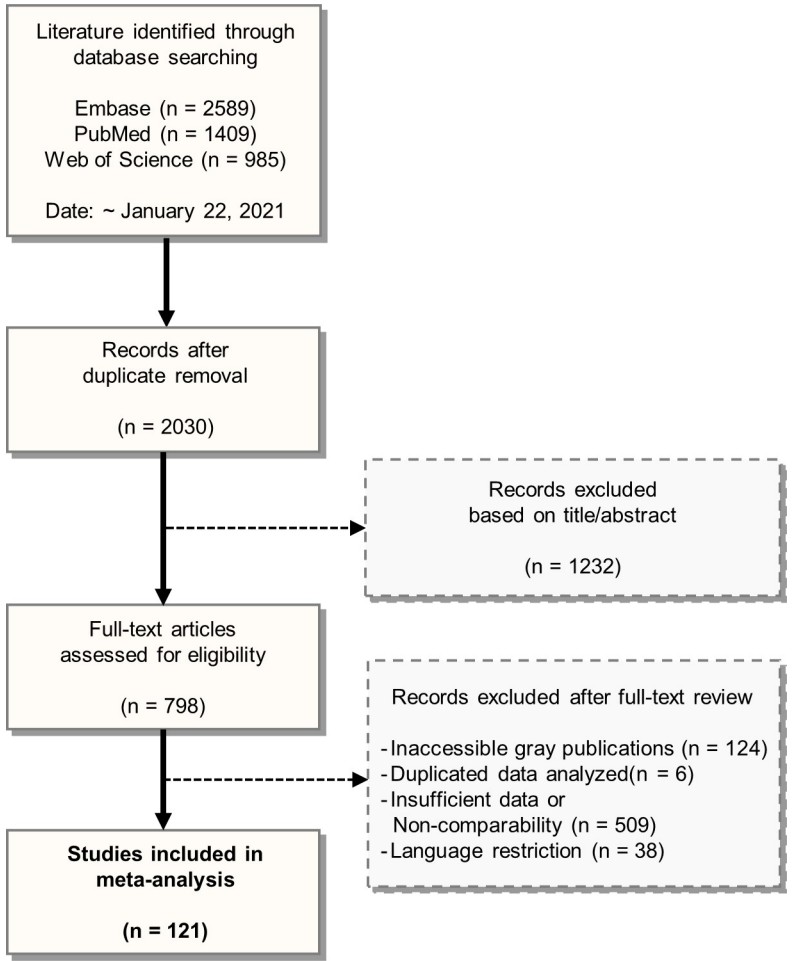

**Fig 1. PRISMA flowchart for study selection.** Preferred Reporting Items for Systematic Reviews and Meta-Analyses (PRISMA) flow diagram of literature search and study selection.

## HCV infection

A total of 86 studies that reported HCV serostatus were meta-analyzed (Fig 2A). HCV infection was significantly more prevalent in the tattooed group than in the non-tattooed group (meta-analyzed OR, 2.89; 95% confidence interval [CI], 2.48–3.37). However, a possible publication bias among the studies that reported HCV serostatus was suggested by the funnel plot (Fig 3A) and Egger's test (*p* = 0.01). Nevertheless, HCV infection was still more prevalent in the tattooed group even after adjustment for publication bias using the trim-and-fill method (adjusted OR, 2.37; 95% CI, 2.04–2.76). Likewise, all the subgroup analyses revealed a statistically significant increase in the prevalence of HCV infection in the tattooed group as compared with the non-tattooed group (Table 2).

## HBV infection

In total, 48 studies that reported HBV serostatus were meta-analyzed (Fig 2B). HBV infection was significantly more prevalent in the tattooed group than in the non-tattooed group (meta-analyzed OR, 1.55; 95% CI, 1.31–1.83).

**Table 1. Summary of the included studies.**

| Variables | N | % | Variables | | | N | % |
|---|---|---|---|---|---|---|---|
| **Countries** | **121** | **100** | **Study populations** | | | **116** | **100** |
| Iran | 23 | 19.0 | General population | | | 36 | 31.0 |
| Brazil | 13 | 10.7 | Hospital patients | | | 24 | 20.7 |
| U.S.A. | 12 | 9.9 | Blood donors | | | 6 | 5.2 |
| India | 9 | 7.4 | Intravenous drug users | | | 25 | 21.6 |
| Australia | 6 | 5.0 | Prisoners | | | 25 | 21.6 |
| Mexico | 5 | 4.1 | **Disease** | | | | |
| Taiwan | 5 | 4.1 | HCV | | | 86 | |
| Thailand | 5 | 4.1 | | Anti-HCV | | 81 | 94.2 |
| Nigeria | 4 | 3.3 | | HCV-RNA | | 5 | 5.8 |
| Pakistan | 3 | 2.5 | HBV | | | 48 | |
| Bosnia | 2 | 1.7 | | HBsAg | | 38 | 79.2 |
| Canada | 2 | 1.7 | | Anti-HBc | | 6 | 12.5 |
| Ethiopia | 2 | 1.7 | | HBV-DNA | | 3 | 6.3 |
| France | 2 | 1.7 | | HBs Ab | | 1 | 2.1 |
| Italy | 2 | 1.7 | HIV | | | 20 | |
| Spain | 2 | 1.7 | | Anti-HIV | | 20 | 100 |
| Others | 24 | 19.8 | Syphilis | | | 3 | |
| | | | | RPR, VDRL test | | 3 | 100 |

Abbreviations: HCV, hepatitis C virus; HBV, hepatitis B virus; HIV, human immunodeficiency virus; RPR, rapid plasma regain; VDRL, venereal disease research laboratory.

Similarly, all the subgroup analyses revealed an increased prevalence of HBV infection in the tattooed group as compared with the non-tattooed group (Table 2). However, a possible publication bias was suggested for the studies that reported HBV serostatus among general population subgroup (Egger's test, $p = 0.04$), hospital patients (Egger's test, $p = 0.04$), and IV drug users (Egger's test, $p = 0.02$). Nevertheless, a tendency toward increased prevalence of HBV infection was observed among general population subgroup (adjusted OR, 1.41; 95% CI, 0.98–2.03) and IV drug users (adjusted OR, 1.46; 95% CI, 1.16–1.83).

## HIV infection and syphilis

A total of 20 studies that reported HIV serostatus were meta-analyzed (Fig 2C). HIV infection was markedly more prevalent in the tattooed group than in the non-tattooed group (meta-analyzed OR, 3.55; 95% CI, 2.34–5.39). However, only the general population (meta-analyzed OR, 2.73; 95% CI, 1.35–5.54) and the prisoners (meta-analyzed OR, 4.29; 95% CI, 3.32–5.54) showed a statistical significance in the subsequent subgroup analyses (Table 2). For syphilis infection, only 3 studies were identified (Fig 2D). Although syphilis tended to be more prevalent in the tattooed group, it did not reach statistical significance (OR, 1.55; 95% CI, 0.76–3.17). In addition, a subgroup analysis for the study population could not be performed because of the insufficient number of studies.

## Discussion

Our systematic review investigated the association between having a tattoo and TTDs. Previous systematic reviews reported a possible positive association between tattooing and each TTD [18]. For HCV infection, two previous studies reported positive associations with having

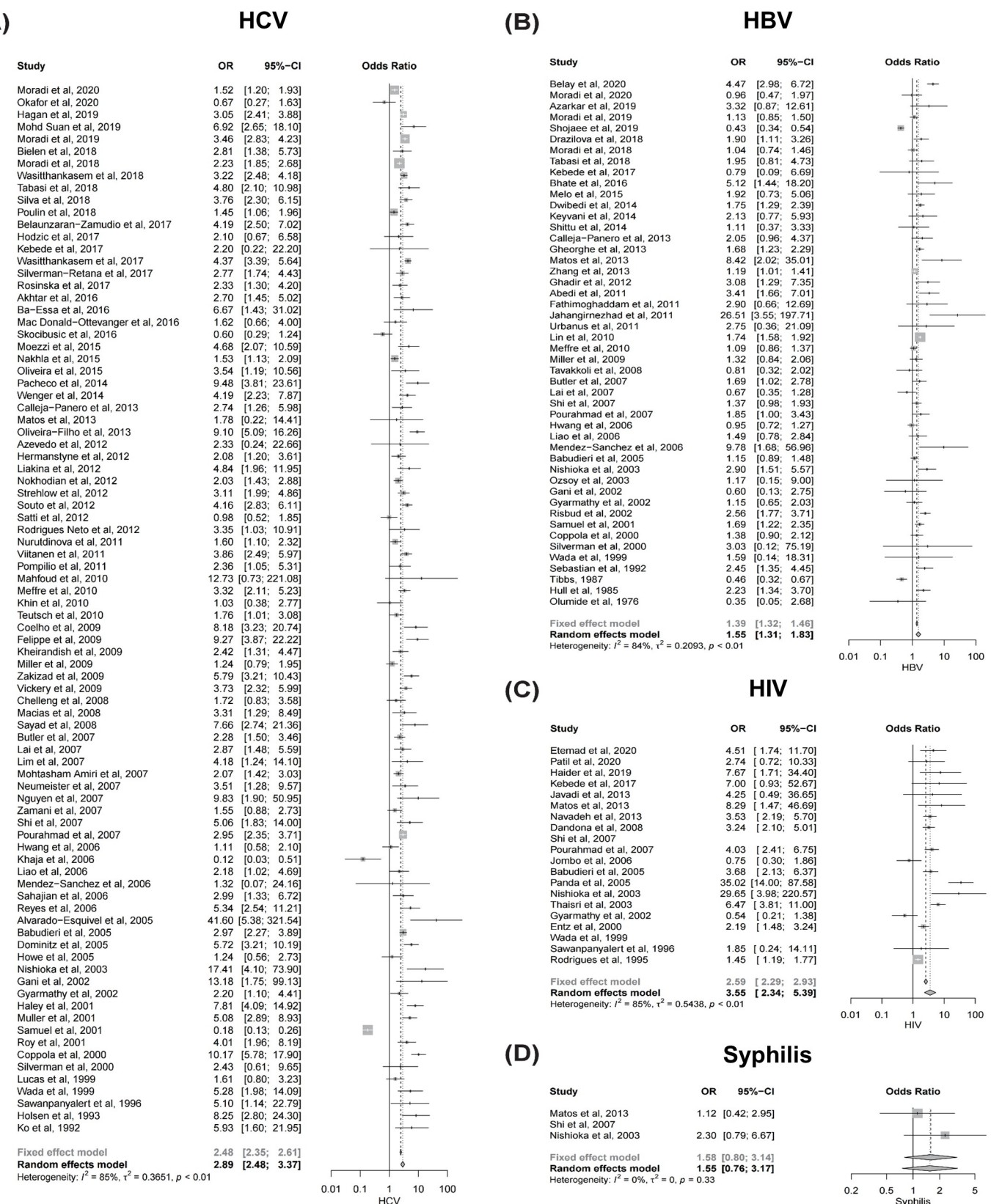

**Fig 2. Forest plots summarizing the meta-analysis.** Forest plots of the meta-analysis. (A) HCV, (B) HBV, (C) HIV, and (D) Syphilis. Abbreviations: HCV, hepatitis C virus; HBV, hepatitis B virus; HIV, human immunodeficiency virus.

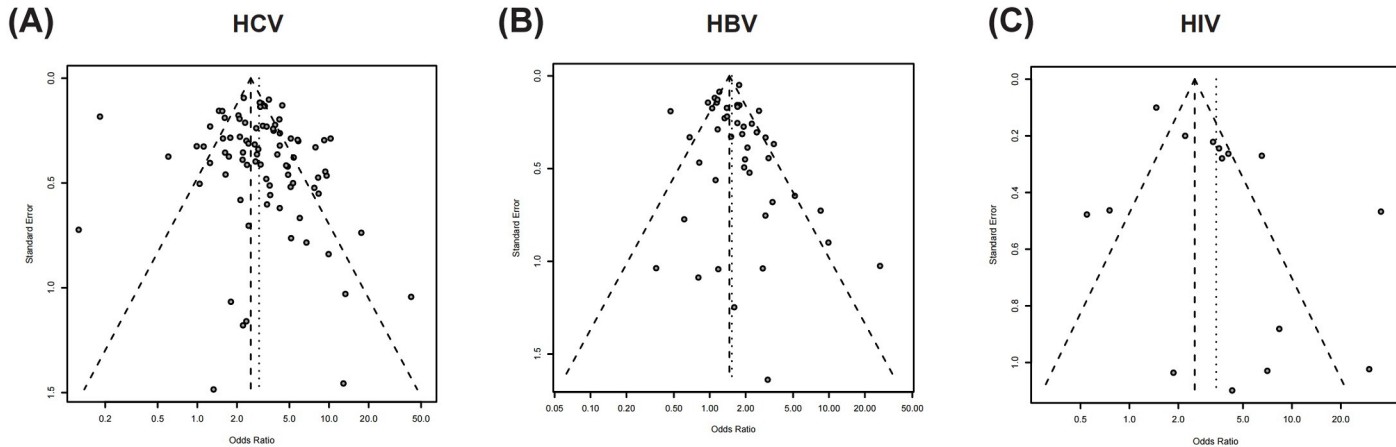

**Fig 3. Funnel plots for the assessment of publication bias.** Funnel plots of the meta-analyses. (A) HCV, (B) HBV, and (C) HIV. Abbreviations: HCV, hepatitis C virus; HBV, hepatitis B virus; HIV, human immunodeficiency virus.

a tattoo, with pooled ORs of 2.74 (95% CI, 2.38–3.15) [19] and 2.79 (95% CI, 2.46–3.18) [17], respectively. In addition, HIV infection has been reported to be more prevalent in tattooed subjects [22]. However, a literature search for syphilis in tattooed subjects is lacking.

**Table 2. Meta-analyzed estimates of the prevalence rates of transfusion-transmitted diseases among subjects with tattoos.**

| Disease | Subgroup[a] | No. of studies | No. of tattoo users | No. of controls | OR (95% CI) | $I^2$ | Egger's test |
|---|---|---|---|---|---|---|---|
| **HCV** | (All) | 86 | 30665 | 171103 | 2.89 (2.48–3.37) | 85% | 0.01 |
| | | | | | [b]2.37 (2.04–2.76) | | |
| | General population | 18 | 5812 | 31492 | 2.94 (2.32–3.73) | 70% | 0.43 |
| | Hospital patients | 16 | 4919 | 46114 | 4.27 (3.21–5.68) | 53% | 0.71 |
| | Blood donors | 5 | 812 | 67477 | 3.27 (1.48–7.21) | 77% | 0.85 |
| | Intravenous drug users | 22 | 3733 | 3501 | 2.37 (1.54–3.67) | 92% | 0.08 |
| | Prisoners | 19 | 12890 | 13275 | 2.99 (2.50–3.56) | 66% | 0.12 |
| **HBV** | (All) | 48 | 25886 | 91495 | 1.55 (1.31–1.83) | 84% | 0.19 |
| | General population | 16 | 5185 | 43828 | 2.04 (1.42–2.93) | 85% | 0.04 |
| | | | | | [b]1.41 (0.98–2.03) | | |
| | Hospital patients | 12 | 2475 | 27141 | 1.97 (1.15–3.37) | 91% | 0.04 |
| | | | | | [b]0.91 (0.54–1.53) | | |
| | Intravenous drug users | 6 | 1276 | 1166 | 1.40 (1.13–1.74) | 0% | 0.02 |
| | | | | | [b]1.46 (1.16–1.83) | | |
| | Prisoners | 9 | 14336 | 13923 | 1.35 (1.07–1.70) | 74% | 0.15 |
| **HIV** | (All) | 20 | 6917 | 17873 | 3.55 (2.34–5.39) | 85% | 0.10 |
| | General population | 5 | 1010 | 6976 | 2.73 (1.35–5.54) | 64% | 0.71 |
| | Hospital patients | 3 | 874 | 2548 | 5.71 (0.87–37.47) | 86% | 0.20 |
| | Intravenous drug users | 3 | 490 | 581 | 4.33 (0.21–88.3) | 95% | 0.67 |
| | Prisoners | 5 | 3712 | 4069 | 4.29 (3.32–5.54) | 0% | 0.21 |
| **Syphilis** | (All) | 3 | 692 | 2203 | 1.55 (0.76–3.17) | 0% | - |

Abbreviations: OR, odds ratio; 95% CI, 95% confidence interval; HCV, hepatitis C virus; HBV, hepatitis B virus; HIV, human immunodeficiency virus

[a]Subgroup analyses were performed when three or more studies were available for each subgroup.

[b]Adjusted OR with the trim-and-fill method for possible publication bias.

In line with previous studies, the results of our study show that the likelihood of having TTDs (HCV, HBV, and HIV infections) among subjects with tattoos is higher than that of the non-tattooed population. This result is consistent with the known knowledge that sharing needles, syringes, or other equipment to inject drugs that may have come in contact with another person's blood is a high-risk factor for TTDs [23–26]. This result was unchanged even after additional adjustment for possible publication bias. Furthermore, we categorized the included studies into TTD subgroups in the general population, hospital patient, blood donor, IV drug user, and prisoner groups to elucidate the specific factors that may have been involved in the spread of TTDs. For HCV infection, all subgroups showed a significant increase in disease transmission rate among the patients with tattoos. However, for HBV infection, only the IV drug user and prisoner groups showed a significant increase in disease incidence rate in the subjects with tattoos after adjusting for publication bias. The prevalence of HIV infection was significantly increased in the general population and prisoner subgroups, although the hospital patients and IV drug users did not show significant differences between the tattooed and non-tattooed groups. This result could be derived from the small number of studies and the included subjects in each subgroup. In the subgroup analyses, the HCV prevalence rate was highest in the hospital subgroup. Although it was statistically not significant, the HIV infection rate was also highest in the hospital subgroup. This may be attributable to the distinguishable characteristic of hospital patients of having awareness of or concern about their illness or symptoms. The IV drug user and prisoner subgroups showed marked increases in the prevalence rates of HCV, HBV, and HIV infections in the tattooed group as compared with the non-tattooed group. Long et al. [27] reported in 2011 that the proportion of prison entrants with tattoos increased with the increasing time spent in the Irish prison over 10 years. They also reported that prisoners who had spent >3 of 10 years were significantly more likely to test positive for HIV antibodies. Adjei et al. [28] also reported that the prevalence rates of HCV, HBV, and HIV infections among prison inmates independently correlated with IV drug use and being incarcerated for >36 months. One study proposed that prisoners tended to be involved in incautious IV drug use because they are exposed to risk behaviors and peer pressures without the concept of sanitation [29]. These results further infer that subjects with tattoos may be associated with increased exposure to unhygienic IV drug use, which leads to the spread of TTDs. Furthermore, a study showed that opioid-substituted treatment in prison led to a reduction in IV injection-associated HIV risk behaviors such as injecting drugs or sharing needles [30]. Overall, although our study did not differentiate prison inmates with tattoos before and after the incarceration, the results imply that prisoners and IV drug users who might have been exposed to an unsafe environment during tattooing could contribute to the increased prevalence of TTDs.

The association of having a tattoo with HBV infection, although some data indicated statistical significance, was not evident compared to the patients with HCV and HIV infections. We assumed that this tendency is attributed to the availability of vaccines for HBV, unlike HCV and HIV, for which vaccines are not yet available. Gahrton et al. [31] reported that the proportion of prisoners in Stockholm who had received HBV vaccination was estimated to be 40.6%. Vaccination prior to the infection may have played a protective role and decreased the overall morbidity of HBV infection. Information on each subject's vaccination history could be useful for evaluating the independent effect of vaccination in future analysis. Moreover, the included studies were predominantly from countries with low HBV prevalence rates [32]. For instance, the largest number of papers and subjects included for HBV infection analysis were from Iran, where the HBV prevalence was <2% [32]. Moreover, among all the included subjects, 59% were from countries with low HBV prevalence rates (<2%), which might have led to reduced statistical power.

For syphilis infection, the prevalence rate was not significantly higher among subjects with tattoos compared to those without tattoos. Although this might be primarily attributed to the small number of studies for the evaluation, several other elements must be considered. First, all the involved studies used screening tests for the diagnosis of syphilis infection. With these non-treponemal tests with high sensitivity [33–35], false-positive cases should be ruled out through confirmatory treponemal tests such as the fluorescent treponemal antibody-absorbed test or *Treponema pallidum* particle agglutination test [25, 33–35]. Second, the characteristic cutaneous manifestations of syphilis may lead to earlier recognition and initiation of interventions that can provide cure and thus cause negative screening test results [33, 34].

Our results raise other possibilities that could lead to an elevated incidence of TTDs in the tattooed population. Drews et al. [36] investigated behavioral differences in tattooed and non-tattooed college students using self-evaluation questionnaires. The tattooed male students' responses showed increased incidence of participation in risky behaviors, presence of more sexual partners, and higher arrest rates. The responses of tattooed females revealed an increased incidence of drug use and body piercings. These behaviors may also have been associated with the increased prevalence of TTDs.

From the perspective of the intrinsic properties of observational studies, this study has several limitations. First, the heterogeneity of the included meta-analyzed studies was considerably high. Second, we did not evaluate other databases (e.g., language-restricted and gray literature), which could have enabled us to access further epidemiological information. In addition, too few studies regarding syphilis in subjects with and without tattoos were available for the meta-analysis. Moreover, other behavioral factors (e.g., multiple sex partners), which are more common in tattooed subjects, might have been confounding factors and led to an overestimation of the effect of tattooing in TTD prevalence. Establishing a temporal relationship between having a tattoo and the morbidity of TTDs is also essential; however, we could not clarify this relationship owing to the lack of information in the included studies. Nevertheless, our study has strengths in that it presents a comprehensive review of a large number of studies covering various TTDs with a consistent methodology, and comprises and summarizes updated studies on the relationship between having tattoos and TTDs.

In conclusion, this study suggests that TTDs are more prevalent in people with tattoos than in those without tattoos. Apart from the hazardous effects of the tattoo materials themselves, the unhygienic conditions in which the procedures are performed may be associated with the spread of TTDs. Our study results support the idea that having a tattoo could be a risk factor for TTDs.

## Supporting information

**S1 File. Search strategies for database.**
(DOCX)

**S1 Table. Characteristics and main findings of the included studies.**
(DOCX)

**S2 Table. Target diseases and related findings of the included studies.**
(DOCX)

**S3 Table. Quality assessment of analytical studies that used the adapted Newcastle-Ottawa scale for cross-sectional studies.**
(DOCX)

**S1 Checklist. PRISMA 2009 checklist.**
(DOC)

## Author Contributions

**Conceptualization:** Joung Soo Kim, Mi Youn Park, Eung Ho Choi.

**Data curation:** Sung Ha Lim, Solam Lee, Young Bin Lee, Chung Hyeok Lee, Jong Won Lee, Sang-Hoon Lee, Ju Yeong Lee.

**Formal analysis:** Sung Ha Lim, Solam Lee.

**Funding acquisition:** Eung Ho Choi.

**Investigation:** Sung Ha Lim, Solam Lee.

**Methodology:** Solam Lee, Sang Baek Koh.

**Project administration:** Eung Ho Choi.

**Resources:** Sung Ha Lim.

**Software:** Sung Ha Lim, Solam Lee.

**Supervision:** Eung Ho Choi.

**Validation:** Joung Soo Kim, Mi Youn Park, Sang Baek Koh, Eung Ho Choi.

**Visualization:** Sung Ha Lim, Solam Lee.

**Writing – original draft:** Sung Ha Lim, Solam Lee.

**Writing – review & editing:** Joung Soo Kim, Mi Youn Park, Eung Ho Choi.

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
