## [Decision Letter · Decision Letter 0]

20 Jul 2021

PONE-D-21-05095

Increased prevalence of transfusion-transmitted diseases among people with tattoos: A systematic review and meta-analysis

PLOS ONE

Dear Dr. Choi,

Thank you for submitting your manuscript to PLOS ONE. After careful consideration, we feel that it has merit but does not fully meet PLOS ONE’s publication criteria as it currently stands. Therefore, we invite you to submit a revised version of the manuscript that addresses the points raised during the review process.

We look forward to receiving your revised manuscript.

Kind regards,

Maria R. Khan, PhD, MPH

Academic Editor

PLOS ONE

Journal Requirements:

3. During our internal review, we noticed you have overlapping text with an abstract of your work published here:

http://www.papersearch.net/thesis/article.asp?key=3845077

PLOS ONE cannot (re)publish material without sufficient permission from the original copyright holder to publish under a CC BY license. Please provide proof that the owner of the content (a) has given you written permission to use it, and (b) has approved of the CC BY license being applied to their content. You may have the following form completed by the owner as proof: https://journals.plos.org/plosone/s/file?id=7c09/content-permission-form.pdf. Alternatively, you may electronically request permissions electronically from the copyright owner and send us proof of approval, as long as the approval clearly shows that the owner has approved of the CC BY license being applied to their content. Please see https://journals.plos.org/plosone/s/licenses-and-copyright for more information.

Reviewers' comments:

Reviewer's Responses to Questions

**Comments to the Author**

1. Is the manuscript technically sound, and do the data support the conclusions?

Reviewer #1: Partly

Reviewer #2: Yes

2. Has the statistical analysis been performed appropriately and rigorously? 

Reviewer #1: I Don't Know

Reviewer #2: Yes

3. Have the authors made all data underlying the findings in their manuscript fully available?

Reviewer #1: Yes

Reviewer #2: Yes

4. Is the manuscript presented in an intelligible fashion and written in standard English?

Reviewer #1: Yes

Reviewer #2: Yes

5. Review Comments to the Author

Reviewer #1: This article appears to be a rigorously conducted literature review. You accessed observational data comparing tattooed populations to non-tattooed populations to determine prevalence of Hepatitis B, C, HIV and syphilis. However, a few of the statements in your conclusion suggest causation, and you have not been able to determine causation via this literature review, only association. For example, on page 12, lines 202-206, you state that: "the incidence of TTDs in the tattooed population is higher than in the general population BECAUSE having or performing tattoos is a modifiable factor for preventing the spread of such diseases...our study indicates an INCREASE IN THE RISK OF TTDs among subjects with tattoos." Finally, at the end of the Discussion section, page 15, lines 274-275, you state: "tattooing procedures performed under unhygienic conditions SEEM TO BE THE CORE MATTER IN THE SPREAD OF TTDs."

I interpret these statements as indicating that tattooing is responsible for the spread of TTDs, but I don't believe the is what your data show. Instead, they show that the likelihood of having a TTD is higher among those with tattoos, but there is no causal evidence in your review. The authors try to address this by doing analyses with different sub-populations, such as prisoners and IV drug users. However, there is no statement in the article that addresses the possibility that other behaviors that may be more common among people with tattoos could be responsible for the spread of TTDs and there was no discussion of the conditions under which the people in these studies were tattooed (ie, there is a huge difference between getting a tattooed from a licensed artist who has received training on prevention of communicable diseases and an untrained person in a prison who is not using sterilized, single-use equipment). In addition, there is no discussion of the timing of the TTD transmission and the tattoo. For these reasons, that statements mentioned above should be reworded to be clear that there was no proof of causation, but only association. I do not see a limitations section either where you could acknowledge that there could be other behaviors that predisposed these individuals to TTDs.

Additionally, it would be useful if you define the term "community-dweller" as it is not immediately clear to the reader who is being described by this term.

Reviewer #2: The authors have conducted a systematic review and meta-analysis to determine the prevalence of transfusion transmitted diseases among people with tattoos. The study adds evidence to increased risk of TTDs associated with tattoos especially HIV. Thank you for the tremendous work.

6. PLOS authors have the option to publish the peer review history of their article (what does this mean?). If published, this will include your full peer review and any attached files.

Reviewer #1: **Yes: **Rebecca Giguere

Reviewer #2: No

---

## [Author Response · Author response to Decision Letter 0]

2 Aug 2021

Review Comments to the Author

Reviewer #1: This article appears to be a rigorously conducted literature review. You accessed observational data comparing tattooed populations to non-tattooed populations to determine prevalence of Hepatitis B, C, HIV and syphilis. 

However, a few of the statements in your conclusion suggest causation, and you have not been able to determine causation via this literature review, only association. For example, on page 12, lines 202-206, you state that: "the incidence of TTDs in the tattooed population is higher than in the general population BECAUSE having or performing tattoos is a modifiable factor for preventing the spread of such diseases...our study indicates an INCREASE IN THE RISK OF TTDs among subjects with tattoos." Finally, at the end of the Discussion section, page 15, lines 274-275, you state: "tattooing procedures performed under unhygienic conditions SEEM TO BE THE CORE MATTER IN THE SPREAD OF TTDs."

I interpret these statements as indicating that tattooing is responsible for the spread of TTDs, but I don't believe the is what your data show. Instead, they show that the likelihood of having a TTD is higher among those with tattoos, but there is no causal evidence in your review. The authors try to address this by doing analyses with different sub-populations, such as prisoners and IV drug users. 

However, there is no statement in the article that addresses the possibility that other behaviors that may be more common among people with tattoos could be responsible for the spread of TTDs and there was no discussion of the conditions under which the people in these studies were tattooed (ie, there is a huge difference between getting a tattooed from a licensed artist who has received training on prevention of communicable diseases and an untrained person in a prison who is not using sterilized, single-use equipment). In addition, there is no discussion of the timing of the TTD transmission and the tattoo. For these reasons, that statements mentioned above should be reworded to be clear that there was no proof of causation, but only association. I do not see a limitations section either where you could acknowledge that there could be other behaviors that predisposed these individuals to TTDs.

Additionally, it would be useful if you define the term "community-dweller" as it is not immediately clear to the reader who is being described by this term.

Response:

 We greatly appreciate your suggestions regarding our manuscript. We have carefully reviewed and updated the text as suggested.

First, we have revised all the sentences that stated or implied a causal relationship between tattooing and TTD infections. The sentence you first pointed out has been deleted. The expressions on page 12, lines 210-212; page 14, lines 242-245; page 16, lines 290-292, have been revised into objective phrases that interpret the results of the data (highlighted in the revised manuscript).

- Page 12, lines 210-212:

In line with previous studies, the results of our study show that the likelihood of having TTDs (HCV, HBV, and HIV infections) among subjects with tattoos is higher than that of the non-tattooed population.

- Page 14, lines 242-245:

Overall, although our study did not differentiate prison inmates with tattoos before and after the incarceration, the results imply that prisoners and IV drug users who might have been exposed to an unsafe environment during tattooing could contribute to the increased prevalence of TTDs.

- Page 16, lines 290-292:

Apart from the hazardous effects of the tattoo materials themselves, the unhygienic conditions in which the procedures are performed may be associated with the spread of TTDs. 

 Second, we strongly agree with your opinion that other factors may influence the incidence of TTDs. Thus, we have addressed possible confounding factors in the revised manuscript. 

 A study has scrutinized tattoo-associated behaviors. Drews et al. [1] investigated behavioral differences in tattooed and non-tattooed college students using self-evaluation questionnaires. The tattooed male students’ responses showed increased incidence of participation in risky behaviors, presence of more sexual partners, and higher arrest rates. The responses of tattooed females revealed an increased incidence of drug use and body piercings. These behaviors may also have been associated with the increased prevalence of TTDs.

 In addition, establishing a temporal relationship between having a tattoo and the morbidity of the TTDs is also essential; however, we could not clarify this relationship owing to the lack of information in the included studies.

We have added the underlined explanation in the Discussion section and also stated this point as a limitation in our revised manuscript (highlighted, page 15, lines 268-274 and lines 280-285). The limitation section is within discussion section, page 15, lines 275-285.

Lastly, we defined “community-dwellers” as those who live within certain geographic areas, in accordance with the studies included (page 7, lines 124-128).

1. Drews DR, Allison CK, Probst JR. Behavioral and self-concept differences in tattooed and nontattooed college students. Psychol Rep. 2000; 86(2):475-81. https://doi.org/10.2466/pr0.2000.86.2.475 PMID:10840898.

---

## [Decision Letter · Decision Letter 1]

27 Sep 2021

PONE-D-21-05095R1Increased prevalence of transfusion-transmitted diseases among people with tattoos: A systematic review and meta-analysisPLOS ONE

Dear Dr. Choi,

Thank you for submitting your manuscript to PLOS ONE. After careful consideration, we feel that it has merit but does not fully meet PLOS ONE’s publication criteria as it currently stands. Therefore, we invite you to submit a revised version of the manuscript that addresses the points raised during the review process. Specifically please see review and revise use of the term "community dweller," which is vague.

Please submit your revised manuscript within four weeks. If you will need more time than this to complete your revisions, please reply to this message or contact the journal office at plosone@plos.org. Please include the following items when submitting your revised manuscript:A rebuttal letter that responds to each point raised by the academic editor and reviewer(s). You should upload this letter as a separate file labeled 'Response to Reviewers'.A marked-up copy of your manuscript that highlights changes made to the original version. You should upload this as a separate file labeled 'Revised Manuscript with Track Changes'.An unmarked version of your revised paper without tracked changes. You should upload this as a separate file labeled 'Manuscript'.If applicable, we recommend that you deposit your laboratory protocols in protocols.io to enhance the reproducibility of your results. Protocols.io assigns your protocol its own identifier (DOI) so that it can be cited independently in the future. For instructions see: https://journals.plos.org/plosone/s/submission-guidelines#loc-laboratory-protocols. Additionally, PLOS ONE offers an option for publishing peer-reviewed Lab Protocol articles, which describe protocols hosted on protocols.io. Read more information on sharing protocols at https://plos.org/protocols?utm_medium=editorial-email&utm_source=authorletters&utm_campaign=protocols.

We look forward to receiving your revised manuscript.

Kind regards,

Maria R. Khan, PhD, MPH

Academic Editor

PLOS ONE

Journal Requirements:

Reviewers' comments:

Reviewer's Responses to Questions

**Comments to the Author**

1. If the authors have adequately addressed your comments raised in a previous round of review and you feel that this manuscript is now acceptable for publication, you may indicate that here to bypass the “Comments to the Author” section, enter your conflict of interest statement in the “Confidential to Editor” section, and submit your "Accept" recommendation.

Reviewer #1: (No Response)

2. Is the manuscript technically sound, and do the data support the conclusions?

Reviewer #1: Yes

3. Has the statistical analysis been performed appropriately and rigorously? 

Reviewer #1: I Don't Know

4. Have the authors made all data underlying the findings in their manuscript fully available?

Reviewer #1: Yes

5. Is the manuscript presented in an intelligible fashion and written in standard English?

Reviewer #1: Yes

6. Review Comments to the Author

Reviewer #1: Thank you for your revisions. You have addressed my comments sufficiently, except that the term "community dweller" is still unclear to me. I believe perhaps you are referring to people who live in urban areas? In which case, you could put urban dwellers or city dwellers? Otherwise, it remains unclear why someone who lives within a certain geographic area would be at higher risk for TTD's, unless you describe some other particular characteristics of those geographic areas.

7. PLOS authors have the option to publish the peer review history of their article (what does this mean?). If published, this will include your full peer review and any attached files.

Reviewer #1: **Yes: **Rebecca Giguere

---

## [Author Response · Author response to Decision Letter 1]

17 Oct 2021

Reviewers' comments:

Reviewer #1: 

Thank you for your revisions. You have addressed my comments sufficiently, except that the term "community dweller" is still unclear to me. I believe perhaps you are referring to people who live in urban areas? In which case, you could put urban dwellers or city dwellers? Otherwise, it remains unclear why someone who lives within a certain geographic area would be at higher risk for TTD's, unless you describe some other particular characteristics of those geographic areas.

Response:

Thank you for your considerate comment.

The term “community-dwellers” was initially defined as people who live within certain geographic areas (urban as well as rural) according to population-based studies. However, we absolutely agree with your comment that this term is vague.

We believe that this community subgroup referred to in population studies could represent the general population, apart from the specific populations and circumstances to which they belong. Other subgroups defined in our study included hospitalized patients, blood donors, intravenous drug users, and prisoners. Such subgroups are within characteristic environments which favor an increased prevalence of TTDs in the tattooed population. Thus, we thought we could evaluate general prevalence by assessing community-based subgroup. Certainly, the community sample was not evenly recruited from all over the world; however, we thought it is worth estimating the prevalence in a sample which can be approximated to the general population.

To convey the meaning of the term properly, we revised the term “community-dwellers” to “general population”.

We added such descriptions and replaced the original term with “general population” in the revised manuscript (Highlighted, lines 126-130)

---

## [Editor Report · Decision Letter 2]

11 Jan 2022

Increased prevalence of transfusion-transmitted diseases among people with tattoos: A systematic review and meta-analysis

PONE-D-21-05095R2

Dear Dr. Choi,

We’re pleased to inform you that your manuscript has been judged scientifically suitable for publication and will be formally accepted for publication once it meets all outstanding technical requirements.

Kind regards,

Maria R. Khan, PhD, MPH

Academic Editor

PLOS ONE
---

## [Editor Report · Acceptance letter]

17 Jan 2022

PONE-D-21-05095R2 

Increased prevalence of transfusion-transmitted diseases among people with tattoos: A systematic review and meta-analysis 

Dear Dr. Choi:

I'm pleased to inform you that your manuscript has been deemed suitable for publication in PLOS ONE. Congratulations! Your manuscript is now with our production department. 

Kind regards, 

on behalf of

Dr. Maria R. Khan 

Academic Editor

PLOS ONE